# Impact of immune evasion, waning and boosting on dynamics of population mixing between a vaccinated majority and unvaccinated minority

David N. Fisman[1]*, Afia Amoako[1], Alison Simmons[1], Ashleigh R. Tuite[1,2]

**1** Dalla Lana School of Public Health, University of Toronto, Toronto, Ontario, Canada, **2** Centre for Immunization Programs, Public Health Agency of Canada, Ottawa, Ontario, Canada

* david.fisman@utoronto.ca

**Data Availability Statement:** The full model is available via the Internet at https://figshare.com/articles/dataset/EXCEL_MODEL_WITH_WANING_FOR_POSTING_xls/21926127.

## Abstract

### Background

We previously demonstrated that when vaccines prevent infection, the dynamics of mixing between vaccinated and unvaccinated sub-populations is such that use of imperfect vaccines markedly decreases risk for vaccinated people, and for the population overall. Risks to vaccinated people accrue disproportionately from contact with unvaccinated people. In the context of the emergence of Omicron SARS-CoV-2 and evolving understanding of SARS-CoV-2 epidemiology, we updated our analysis to evaluate whether our earlier conclusions remained valid.

### Methods

We modified a previously published Susceptible-Infectious-Recovered (SIR) compartmental model of SARS-CoV-2 with two connected sub-populations: vaccinated and unvaccinated, with non-random mixing between groups. Our expanded model incorporates diminished vaccine efficacy for preventing infection with the emergence of Omicron SARS-CoV-2 variants, waning immunity, the impact of prior immune experience on infectivity, "hybrid" effects of infection in previously vaccinated individuals, and booster vaccination. We evaluated the dynamics of an epidemic within each subgroup and in the overall population over a 10-year time horizon.

### Results

Even with vaccine efficacy as low as 20%, and in the presence of waning immunity, the incidence of COVID-19 in the vaccinated subpopulation was lower than that among the unvaccinated population across the full 10-year time horizon. The cumulative risk of infection was 3–4 fold higher among unvaccinated people than among vaccinated people, and unvaccinated people contributed to infection risk among vaccinated individuals at twice the rate that would have been expected based on the frequency of contacts. These findings were robust across a range of assumptions around the rate of waning immunity, the impact of "hybrid

immunity", frequency of boosting, and the impact of prior infection on infectivity in unvaccinated people.

## Interpretation

Although the emergence of the Omicron variants of SARS-CoV-2 has diminished the protective effects of vaccination against infection with SARS-CoV-2, updating our earlier model to incorporate loss of immunity, diminished vaccine efficacy and a longer time horizon, does not qualitatively change our earlier conclusions. Vaccination against SARS-CoV-2 continues to diminish the risk of infection among vaccinated people and in the population as a whole. By contrast, the risk of infection among vaccinated people accrues disproportionately from contact with unvaccinated people.

## Introduction

The rapid development of safe and effective vaccines was a sentinel achievement of the SARS-CoV-2 pandemic and has likely prevented millions of deaths globally [1, 2]. However, the use of vaccine mandates as a means of encouraging vaccine uptake has proven controversial, with opponents suggesting that vaccination requirements for work, school or travel represent unreasonable restrictions of individual rights [3]. We previously used a simple mathematical model of disease transmission and vaccine effect, as well as non-random population mixing to explore how vaccination, and different mixing patterns between vaccinated and unvaccinated populations would affect risk and disease dynamics for each sub-population [4]. In this work, we created a metric of the disproportionate impact of infection from unvaccinated sub-populations on risk among vaccinated people when vaccines are imperfect [4].

We found that the risk of infection was markedly higher among unvaccinated people than among vaccinated people for all assumptions about mixing between the two groups, even with lower-efficacy vaccines (VE ~ 40%) [4]. We also found that the contact-adjusted contribution of unvaccinated people to infection risk was disproportionate, with unvaccinated people contributing to infections among those who were vaccinated at a rate higher than would have been expected based on contact numbers alone [4]. Finally, we found that as like-with-like mixing increased (with vaccinated and unvaccinated people interacting preferentially with those of similar vaccination status), attack rates among vaccinated people decreased and attack rates among unvaccinated people increased, but the contact-adjusted contribution to risk among vaccinated people derived from contact with unvaccinated people increased [4]. This led us to suggest that while risk associated with avoiding vaccination during a virulent pandemic accrues chiefly to people who are unvaccinated, their choices affect risk of viral infection among those who are vaccinated in a manner that is disproportionate to the portion of unvaccinated people in the population. Implicitly then, this model supported the use of vaccine mandates.

Our publication was met with some criticism, some scientific and some that could be characterized as more ideological. We responded to scientific criticism in a published response [5]. Most criticism focused on the diminished vaccine efficacy associated with emergence of the Omicron variant, the fact that we had assumed durable immunity from vaccination in our published model, and the notion that giving unvaccinated people a "head start" of only 20% baseline immunity was insufficient. Evolving information on vaccine efficacy [6–8], durability

of immune protection provided by vaccination and/or infection [9–13], impacts of vaccination and/or prior infection on infectivity among people with subsequent infection [14, 15], and the availability and effects of booster doses [16, 17], led us to update this earlier work. Our objectives were to evaluate whether the changing understanding of the attributes of SARS-CoV-2 vaccines and variants, and the availability of booster vaccination would result in a qualitative change in our earlier findings in projections using longer time horizons.

## Methods

### Model

Our earlier compartmental model is described in [4]. That model was a compartmental model of a respiratory viral disease with the population subdivided into three possible states: susceptible to infection (S), infected and infectious (I), and recovered from infection with immunity (R). The earlier model was further subdivided to reflect two interconnected sub-populations: vaccinated and unvaccinated. Our revised model was updated to incorporate immune experience related to infection as well as waning immunity, and repeated booster vaccination among the vaccinated population (**Supplementary Appendix Figure 1**). The S, I, and R compartments were divided according to the presence of prior immune experience due to infection. Model equations are presented in the **S1 Appendix**.

Immunity following vaccination was treated as an all-or-none phenomenon, with only a fraction of vaccinated people (as defined by initial vaccine efficacy) entering the model in the immune state and the remainder left in the susceptible state. The emergence of the Omicron variants of concern in late 2021 significantly diminished the efficacy of vaccines against infection with SARS-CoV-2, though there is a broad range of estimates respecting what initial protective efficacy might be. Estimates of 40–64% are most plausible [6–8]; we conservatively used 40% in our base case, and varied initial efficacy across a range of 20–80% in sensitivity analyses. Newer analyses of SARS-CoV-2 vaccines updated to provide protection against XBB viral variant used a test-negative case-control design, and initial efficacy against infection of 50%, which is again consistent with our earlier estimates [18].

Initial immunity after infection was assumed perfect, with all infected people transiting to an immune state upon recovery from infection. The duration of immune protection varied according to vaccination status, as well as prior infection status among vaccinated individuals [11, 19, 20]. Anti-spike antibody titres have been demonstrated to be a consistent correlate of protection against SARS-CoV-2 infection [21]. Townsend et al. have noted that duration of protection seems to be a function of the initial peak antibody titres attained after either infection or vaccination [9, 11] and on that basis have suggested that protection after 2-dose mRNA vaccination is more durable than protection after either natural infection or viral vector vaccination. We used Townshend's estimates to derive a relative hazard of loss of immune protection among vaccinated people relative to loss of immunity after infection in unvaccinated people (**Table 1**).

The occurrence of breakthrough infection in people with prior immunization also appears to confer more durable immune protection (so-called "hybrid immunity") [12, 13, 22], and this was again modelled as a reduced hazard of loss of immunity using data from [13, 20, 22]. The immune correlate of this phenomenon has been demonstrated by Planas et al. [20], and Hoffman et al. [13]. Planas demonstrated that neutralizing titres against Omicron variants seem to have fallen below protective levels at around 5 months following vaccination, but with breakthrough infection titres rose and remained elevated for the entire period available to them for analysis (at least 6 months) [20]. Hoffman demonstrated similar boosting of neutralizing antibody titres in fully vaccinated individuals after breakthrough infection [13]. A recent

**Table 1. Model parameter estimates.**

| Parameter description | Symbol | Value | Plausible Range | Reference |
|---|---|---|---|---|
| Probability of transmission per contact multiplied by contacts per year | β | 728 | 164–728 | Calculated |
| Rate of recovery from infection (per year) | γ | 73 | 41–91 | [38] |
| Basic reproduction number ($R_0$) | $R_0$ | 10 | 6–12 | [39–41] |
| Mixing between subpopulations (0 = random, 1 = assortative) | η | 0.5 | 0–0.9 | Assumption, approach based on [24] |
| Proportion vaccinated | $P_v$ | 0.8 | — | [42] |
| Vaccine efficacy | VE | 0.4 | 0.2–0.8 | [6–8] |
| Approximate population of a Canadian province or region (N) | N | 10,000,000 | — | [43] |
| Mean duration of immune protection from infection (months) | 1/ζ | 10 | 4–16 | [9, 10] |
| Hazard ratio* for loss of immunity with vaccination | $HR_{1V}$ | 0.75 | 0.5–1.0 | [11, 19] |
| Hazard ratio* for loss of immunity with vaccination and prior infection | $HR_{2V}$ | 0.75 | 0.5–1.0 | [20, 22] |
| Reduction in Infectivity (%) | | | | |
| Vaccinated | | 20 | 0–20 | [23, 44] |
| Infected | | 20 | 0–20 | |
| Infected After Vaccination | | 40 | 0–40 | |
| Frequency of Boosting | | 12 | 2–24 | [16, 17] |

*Hazard ratios treated as multiplicative. Thus for vaccinated individuals duration of immune protection is $1/(z*HR_{1V})$; for vaccinated individuals with a history of prior infection duration is $1/(\zeta*HR_{1V}*HR_{2V})$.

systematic review and meta-regression demonstrated that the hazard of loss of immunity after infection in unvaccinated individuals was approximately double that seen after infection in vaccinated people with breakthrough infection [22].

Infectivity of infected individuals was reduced based on vaccination status, prior infection status, or a combination of these two factors [14, 23]. Both immunization and prior infection have been shown to reduce infectivity among people with SARS-CoV-2 infection by approximately 20%, in both household and institutional outbreak settings [14, 15]. As with immune protection, the greatest reduction in infectivity appears to occur in the setting of "hybrid immunity" related to a combination of immunization and prior infection, with infectivity reduced by approximately 40% [14].

We modelled booster vaccination using a periodic step function, which cycled vaccinated but susceptible people back into a vaccine-derived immune state, with a probability reflective of initial vaccine efficacy, as above. Boosting occurred at a frequency of every 2 to 24 months [16, 17], and we assumed that the protective efficacy of vaccination after boosting was the same as the protective efficacy after an initial complete vaccination series.

Mixing between vaccinated and unvaccinated sub-populations was modelled as in our earlier work [4], based on the approach described by Garnett and Anderson [24], with moderate assortativity (like-with-like mixing) used in the base case, and either random mixing or extreme like-with-like mixing evaluated in sensitivity analyses. As in our earlier model, assortativity is determined by a constant, denoted η, with random mixing occurring when η = 0, complete assortativity occurring when η = 1, and intermediate degrees of assortativity occurring at other intermediate values. The fraction of contacts from within a given group, or from an external group, is thus a function of η, as well as the respective sizes of the groups being modeled.

## Analyses

Our base case model was otherwise parameterized to represent a disease similar to SARS-CoV-2 infection with Omicron variants, with an $R_0$ (the reproduction number of an infectious

disease in the absence of immunity or control) of 10 in our base case, consistent with the highly transmissible nature of the Omicron variant [25]. Our model was run over a 10-year time horizon, which was sufficiently long to permit epidemic dynamics to reach equilibrium, but also sufficiently short to be of relevance to decision-makers.

We evaluated the absolute contribution to overall case counts by the vaccinated and unvaccinated sub-populations, as well as within-group and overall infection risk. We estimated both incidence rate ratios for the unvaccinated subpopulation relative to the vaccinated subpopulation, and risk ratios, defined as the ratio of cumulative incidence among the unvaccinated population over the 10-year time horizon, divided by cumulative incidence among the vaccinated population over that same time period. As in our earlier work we estimate a quantity that we denote $\psi$, defined as the incidence of infections among the vaccinated population derived from contact with unvaccinated people, divided by the fraction of the population that is unvaccinated. We estimated $\psi$ both as a time-varying quantity and based on cumulative incidence of infection.

We used the model to explore the impact of varying rates of immunization, varying booster frequency, vaccine efficacy, disease natural history (e.g., basic reproduction numbers) and different levels of like-with-like mixing on the dynamics of disease in vaccinated and unvaccinated sub-populations in sensitivity analyses. We also explored the sensitivity of our results to varying assumptions about the protective effects of prior infection in vaccinated and unvaccinated subpopulations. For the purposes of sensitivity analyses, our outcomes of interest were the risk ratio for infection at 10 years among unvaccinated people, and the average value for $\psi$ over the 10-year time horizon. A working version of our model in Microsoft Excel is available at [26].

## Results

The incidence curve for a simulated epidemic using base case parameters, and with moderate like-with-like mixing ($\eta = 0.5$), 40% initial vaccine efficacy, and 80% vaccination uptake, is presented in **Fig 1**. A large initial pandemic wave was followed by endemic circulation of disease. A majority of the population was vaccinated, and consequently, most cases occurred in vaccinated people, but population-adjusted risk of infection was higher in unvaccinated people over the entire simulated 10-year time period.

Due to the explosive nature of the epidemic among unvaccinated people, the incidence rate ratio for unvaccinated people fell transiently below 1 in the unvaccinated population early on, but quickly rebounded past 1; as the disease reached a stable equilibrium incidence rate ratio among the unvaccinated population remained steady at around 4. The risk ratio for infection among the unvaccinated population (based on cumulative incidence of infection) remained above 1 for the entire 10-year time period, stabilizing around 3.8 (**Fig 2**).

The quantity $\psi$ oscillated over the 10-year time period (**Fig 3**), reflecting both the impact of disease dynamics and periodic boosting, but remained above 1 throughout, signifying a disproportionate contribution to infection risk to vaccinated people by the unvaccinated population. When we estimated $\psi$ cumulatively, the value at 10 years was approximately 2.14, meaning that infection among vaccinated people was more than twice as likely to have been acquired from unvaccinated people than would have been expected based on contact rates alone.

In univariable sensitivity analyses on disease natural history, vaccine efficacy and durability of response, and booster dose frequency, we found no change in qualitative model projections when parameter inputs were varied over plausible ranges. We explored the impact of varying the assortativity constant $\eta$ across a range of values, from random mixing to near-complete

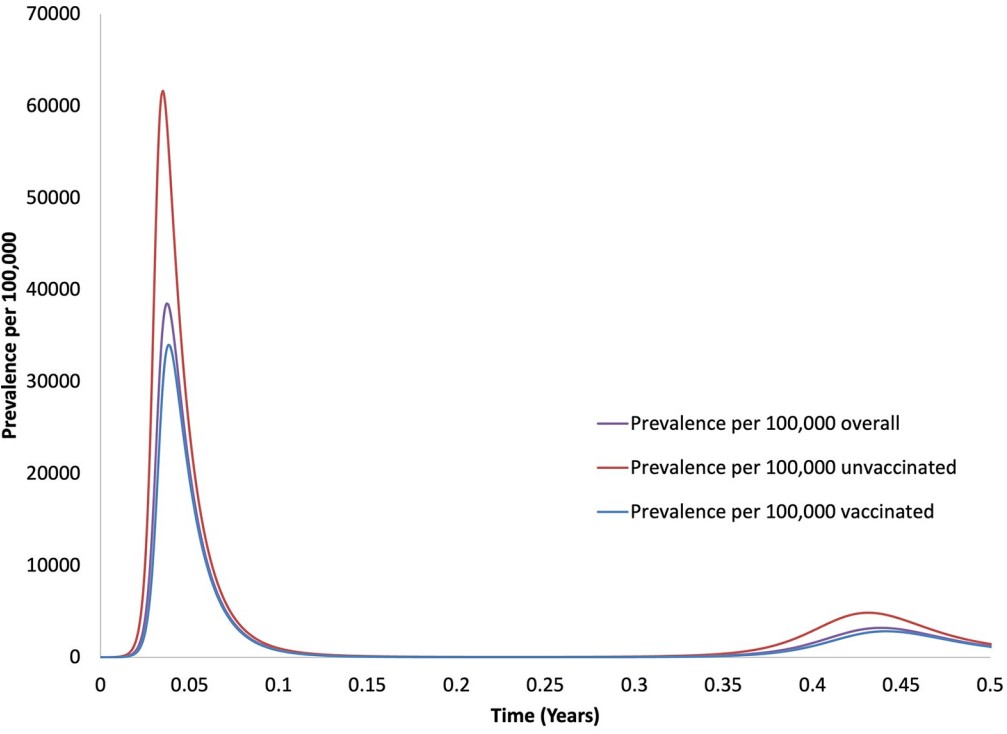

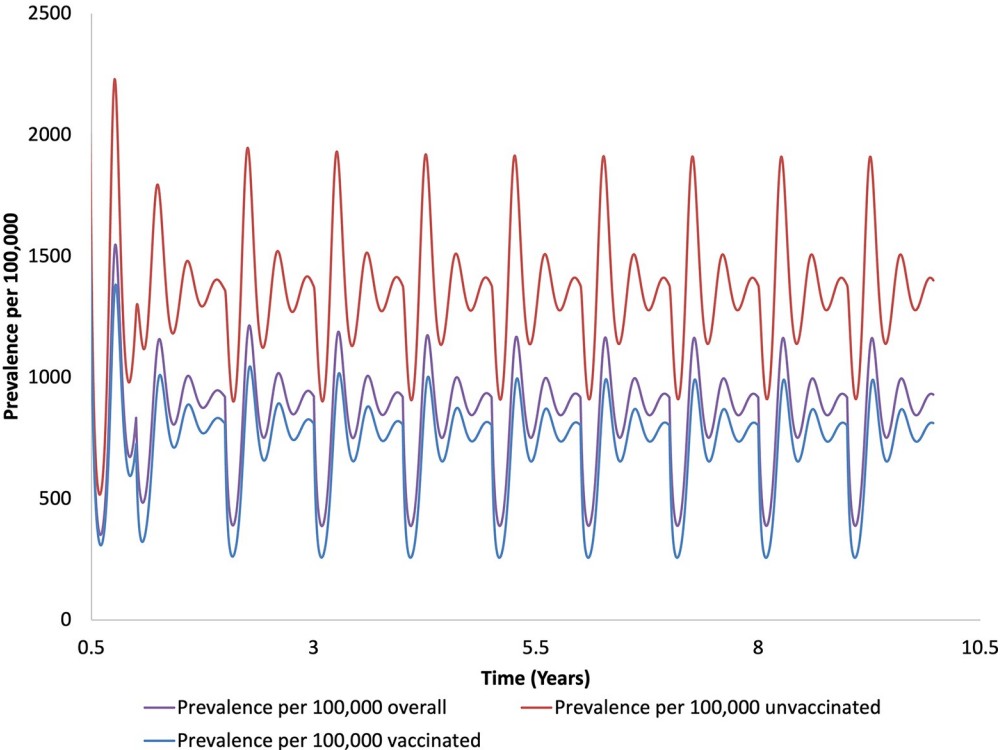

**Fig 1. Simulated epidemic curve for a simulated population and for vaccinated and unvaccinated subpopulations.**
Epidemic curves for the simulated population (purple), and for vaccinated (blue) and unvaccinated (red)
subpopulations. Initial emergence is in panel (A) (0 to 0.5 years) and subsequent endemicity (0.5 to 10 years) is in panel
(B). Periodic oscillation reflects boosting at 1-year intervals. Incidence is highest among unvaccinated people and
lowest among vaccinated people across the 10-year time horizon. Note difference in scales on X- and Y-axes.

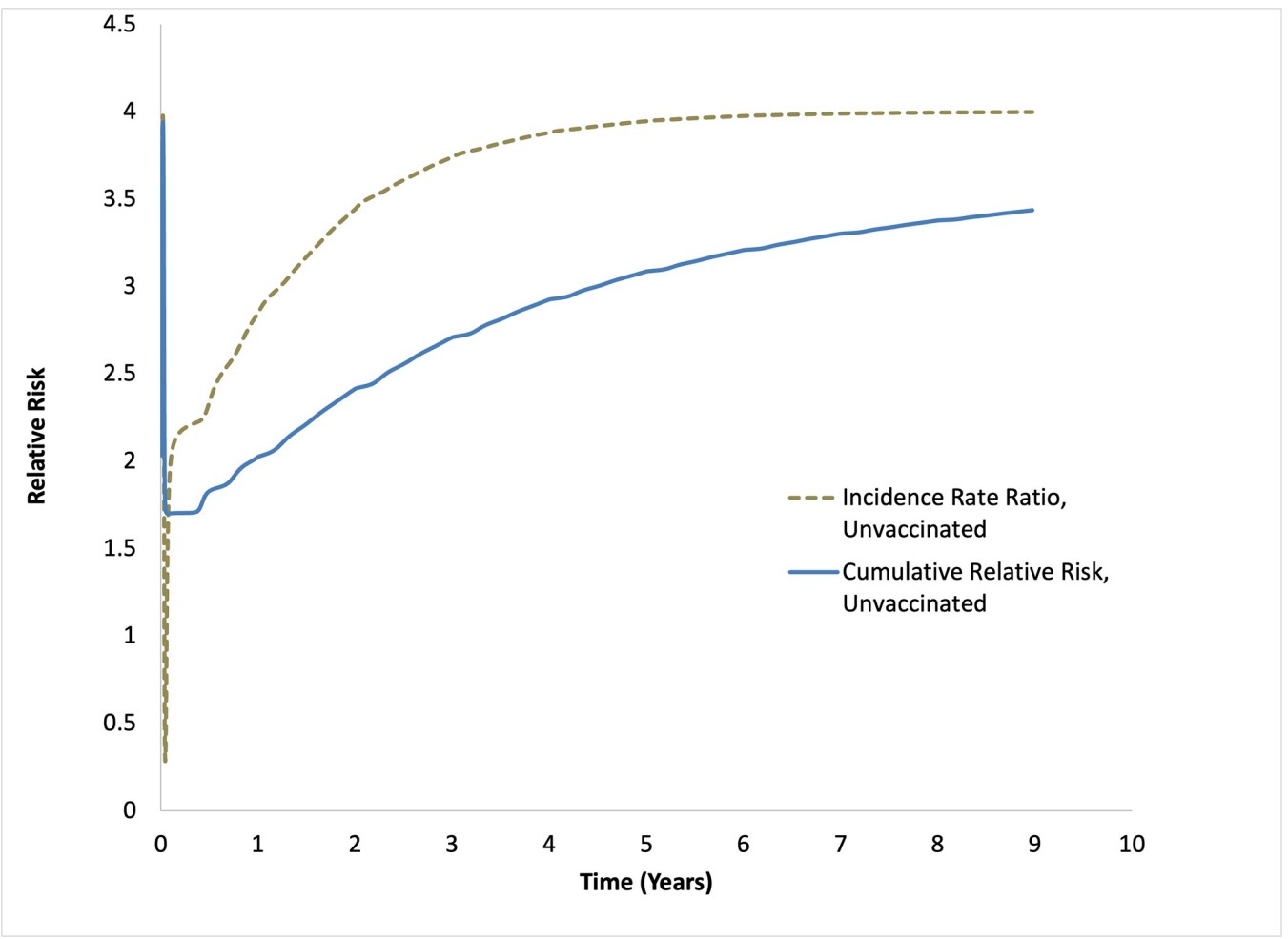

**Fig 2. Relative risk of infection among unvaccinated people.** Relative risk among unvaccinated people is plotted as an incidence rate ratio (dashed gray curve) and as a ratio of cumulative incidence over time (solid blue curve).

like-with-like mixing, and with variation in estimated vaccine efficacy (**Fig 4**). Cumulative $\psi$ rose as like-with-like mixing increased, but was elevated across all scenarios, indicating disproportionate contribution to risk among the vaccinated from the unvaccinated group. By contrast, cumulative relative risk of infection was high (around 3.8) but remained quite stable as assortativity was varied.

The cumulative value of $\psi$ decreased as the relative duration of immune protection after immunization decreased but remained elevated (at 1.4) even when there was no difference in duration of protection between immunity derived from infection, vaccination, and vaccination plus infection (**Supplementary Appendix Figure 2**). The relationship between $\psi$ and boosting frequency was non-linear, likely reflecting interplay between direct protection of vaccinated people and indirect protection of the population as a whole when frequency of boosting was high, but no qualitative differences were seen in projections as boosting frequency was varied from every 2 months to every 24 months (**Supplementary Appendix Figure 3**). Qualitatively, model projections were robust to variation in the impacts of prior infection and vaccination on infectivity, with elevated values of $\psi$ (1.27) even in the unlikely scenario where infection and prior vaccination without infection reduced infectivity by 30%, but infection with prior

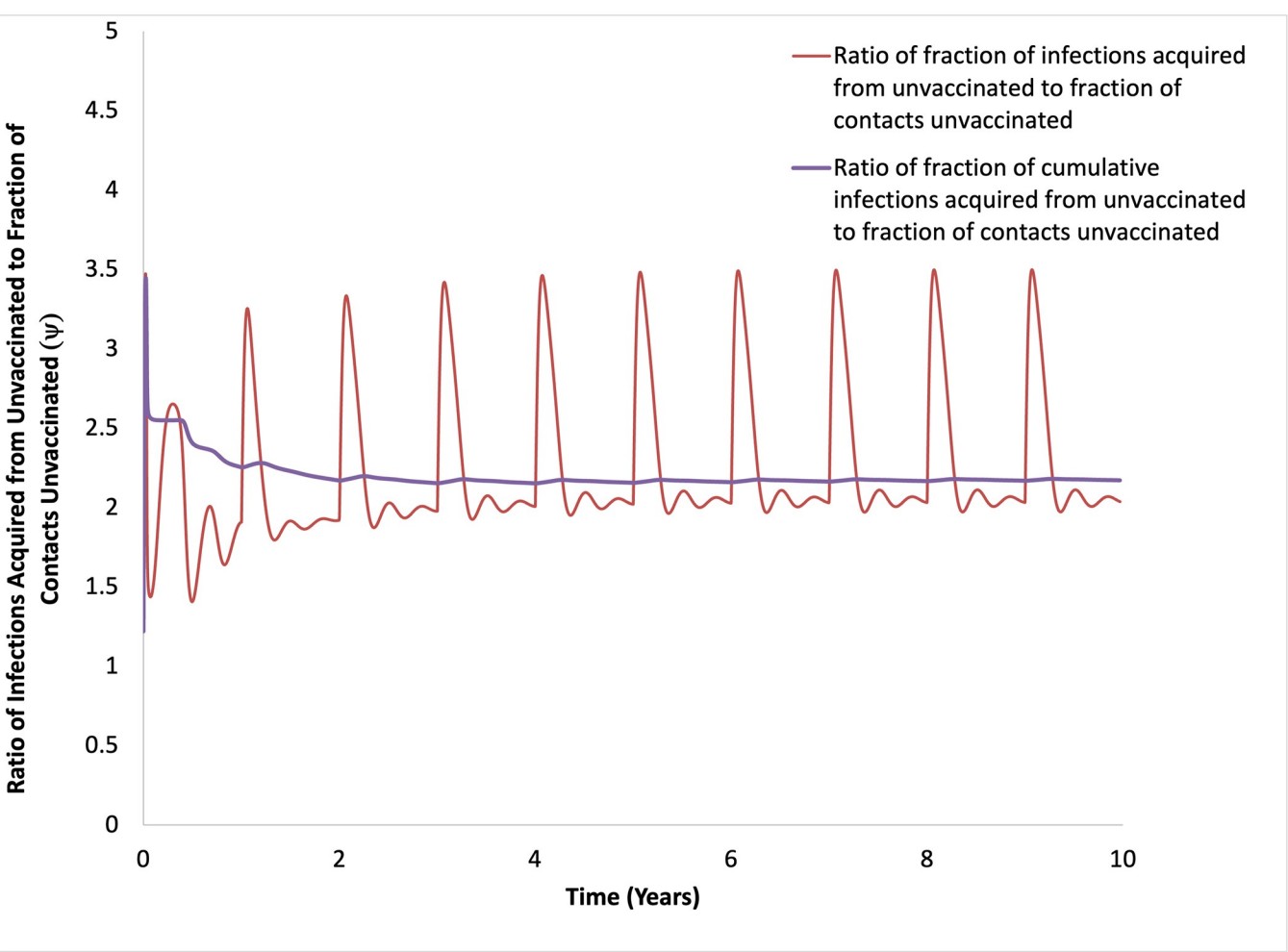

**Fig 3. Disproportionate contribution to infection by unvacccinated people.** Base-case model results plotting the instantaneous value of the quantity ψ (red curve) over time, as well as the cumulative value of ψ (purple curve). ψ is the ratio of the fraction infections acquired by vaccinated people from unvaccinated people divided by fraction of contacts with unvaccinated people. Values > 1 indicate that the contribution to infection risk among vaccinated people from unvaccinated people is disproportionate to contact numbers.

vaccination did not reduce infectivity at all (**Supplementary Appendix Figure 4**). Cumulative relative risks over the 10-year time horizon were far less sensitive to plausible variation in model parameters than cumulative ψ.

## Discussion

Evolving information on Omicron variant SARS-CoV-2 vaccine efficacy, the effects of "hybrid immunity" in vaccinated individuals experiencing breakthrough infection, durability of protection from vaccination and infection, impacts of vaccination and prior infection on infectivity among individuals with subsequent infection, and the availability and effects of booster doses, led us to update an earlier model [4] investigating the impact of mixing between vaccinated and unvaccinated subpopulations. We find that notwithstanding these differences, the basic conclusions from our earlier work remain robust. That is, we find that even with imperfect vaccines, with efficacy as low as 20%, and which wane in protective efficacy over time, vaccination provides direct benefits to vaccinated people, while their infection risk accrues disproportionately from interactions with unvaccinated people. These basic findings remain

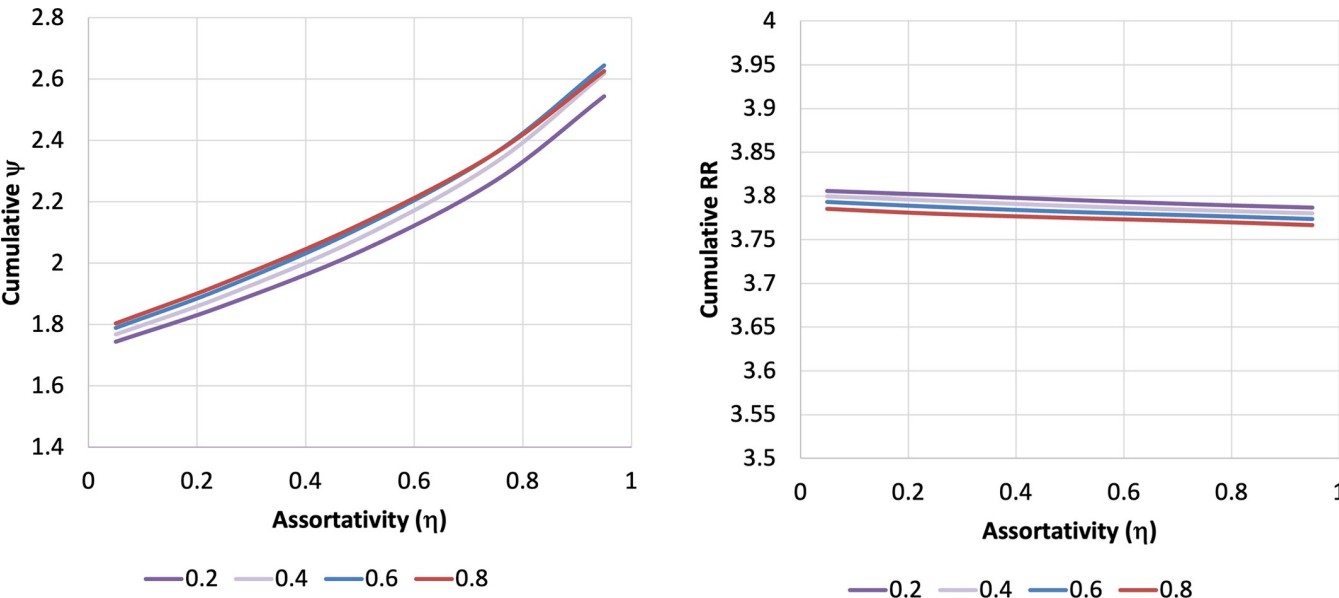

**Fig 4. Assortativity and contribution to risk.** Plots of cumulative value of ψ (A) and cumulative relative risk of infection among the unvaccinated (B) with variation in assortativity (like-with-like mixing). Assortativity (η) is plotted on the X-axes; η = 0 represents random mixing, while higher values of η represent increasing like-with-like mixing. Colored curves represent different values for initial vaccine efficacy (VE), ranging from 0.2 (20%) to 0.8 (80%).

in the face of wide-ranging sensitivity analyses, and over a 10-year time horizon during which the disease moves from an epidemic to endemic state. Furthermore, our findings are consistent with several network-based analyses published in the physical sciences literature, which find that mixing between vaccinated and unvaccinated groups can have important influences on population-level disease dynamics [27, 28]. The importance of assortativity has also been noted in determining the population-level effectiveness of non-pharmaceutical measures such as masking [29, 30].

Indeed, inasmuch as breakthrough infection among vaccinated people is common in our model due to waning immunity and imperfect vaccine efficacy, the reported increase in duration of protection and reduction in infectivity after breakthrough infection in vaccinated people make vaccination extremely impactful on epidemic dynamics notwithstanding the frequency of breakthrough.

In this context, vaccination serves as a kind of immunological priming that occurs without accompanying infectivity that occurs with initial infection rather than vaccination. The advantages that accrue after breakthrough infection may be due to development of mucosal IgA antibody, which could result in a degree of resistance to reinfection due to the presence of this antibody in the upper airway [31]. Vaccination in the absence of prior infection appears to generate low titres of upper airway IgA, whereas individuals with prior infection who receive mRNA vaccine develop higher titres of upper airway IgA [32]. As a matter of policy, encouraging individuals to acquire infection with a virulent pathogen with known tropism for brain [33, 34], blood vessels and the cardiac system [35–37] in pursuit of mucosal immunity is inadvisable, both because this approach generates risk for the individual themselves, and because these individuals become sources of infection for others. Nonetheless, it increasingly appears that prior vaccination combined with unintended breakthrough infection results in important downstream immunological protections for individuals "primed" with mRNA vaccines.

Given the evolving nature of the SARS-CoV-2 pandemic, and of our understanding of the epidemiology of infection and immunity, our analysis is inevitably limited by uncertainty.

However, we have conservatively biased our analyses against vaccines by assuming that the immune response that follows infection is initially perfectly protective against subsequent infection but wanes over time. We do reduce the rate at which protective immunity wanes following breakthrough infection, but we do so in a conservative manner, reducing the rate of waning in the presence of hybrid immunity by only 25%, whereas Planas et al. appear to demonstrate far more profound reductions in waning [20].

In summary, notwithstanding the evolving epidemiology of SARS-CoV-2 infection and understanding of vaccination, we find that incorporating waning immunity, hybrid effects of vaccination and infection, boosting, and a longer time horizon into our earlier model result in no change in our earlier conclusions: that is, that vaccination with currently available vaccines against SARS-CoV-2 results in markedly lower risk of infection over time among vaccinated individuals, while the contact-adjusted risk to vaccinated individuals associated with contact with unvaccinated groups is disproportionate.

## Supporting information

**S1 Appendix.**
(DOCX)

## Author Contributions

**Conceptualization:** David N. Fisman, Afia Amoako, Ashleigh R. Tuite.

**Funding acquisition:** David N. Fisman.

**Methodology:** David N. Fisman, Ashleigh R. Tuite.

**Visualization:** David N. Fisman, Ashleigh R. Tuite.

**Writing – original draft:** David N. Fisman, Afia Amoako, Alison Simmons, Ashleigh R. Tuite.

**Writing – review & editing:** David N. Fisman, Afia Amoako, Alison Simmons, Ashleigh R. Tuite.

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
