## [Decision Letter · Decision Letter 0]

6 Nov 2023

PONE-D-23-24559Impact of Immune Evasion, Waning and Boosting on Dynamics of Population Mixing Between a Vaccinated Majority and Unvaccinated MinorityPLOS ONE

Dear Dr. Fisman,

Thank you for submitting your manuscript to PLOS ONE. After careful consideration, we feel that it has merit but does not fully meet PLOS ONE’s publication criteria as it currently stands. Therefore, we invite you to submit a revised version of the manuscript that addresses the points raised during the review process.

Please submit your revised manuscript by Dec 21 2023 11:59PM. If you will need more time than this to complete your revisions, please reply to this message or contact the journal office at plosone@plos.org. Please include the following items when submitting your revised manuscript:A rebuttal letter that responds to each point raised by the academic editor and reviewer(s). You should upload this letter as a separate file labeled 'Response to Reviewers'.A marked-up copy of your manuscript that highlights changes made to the original version. You should upload this as a separate file labeled 'Revised Manuscript with Track Changes'.An unmarked version of your revised paper without tracked changes. You should upload this as a separate file labeled 'Manuscript'.If applicable, we recommend that you deposit your laboratory protocols in protocols.io to enhance the reproducibility of your results. Protocols.io assigns your protocol its own identifier (DOI) so that it can be cited independently in the future. For instructions see: https://journals.plos.org/plosone/s/submission-guidelines#loc-laboratory-protocols. Additionally, PLOS ONE offers an option for publishing peer-reviewed Lab Protocol articles, which describe protocols hosted on protocols.io. Read more information on sharing protocols at https://plos.org/protocols?utm_medium=editorial-email&utm_source=authorletters&utm_campaign=protocols.

We look forward to receiving your revised manuscript.

Kind regards,

Shinya Tsuzuki, MD, PhD

Academic Editor

PLOS ONE

Journal Requirements:

Additional Editor Comments:

Both reviewers gave positive comments to the manuscript and I agree with their points.

Reviewers' comments:

Reviewer's Responses to Questions

**Comments to the Author**

1. Is the manuscript technically sound, and do the data support the conclusions?

Reviewer #1: Yes

Reviewer #2: Yes

2. Has the statistical analysis been performed appropriately and rigorously? 

Reviewer #1: Yes

Reviewer #2: Yes

3. Have the authors made all data underlying the findings in their manuscript fully available?

Reviewer #1: Yes

Reviewer #2: Yes

4. Is the manuscript presented in an intelligible fashion and written in standard English?

Reviewer #1: Yes

Reviewer #2: Yes

5. Review Comments to the Author

Reviewer #1: The link to your excel model is currently dead, but if that's just a publication/journal issue then that wouldn't be a problem.

I'd recommend changing the orange lines in fig 1 to a darker red, because to many people with the more common types of colour-blindness it looks indistinguishable from the green lines; may also want to ensure fig 1A also has the full legend while you're there. The other orange/green lines can be figured out more easily from context but you might want to think about those too.

Otherwise I can see no flaws; much easier to read than a lot of papers I've seen in the field.

Reviewer #2: The submitted article serves as a follow-up to the authors' previous paper, Fisman, Amoako, Tuite, 'Impact of Population Mixing between Vaccinated and Unvaccinated Subpopulations on Infectious Disease Dynamics: Implications for SARS-CoV-2 Transmission,' which was published in the Canadian Medical Association Journal in April 2022 (https://doi.org/10.1503/cmaj.212105). While the CMAJ paper received comments, including some unjustified criticism, one valid remark was the authors' failure to consider waning immunity in their model. In this submitted paper, they have addressed this concern. While not groundbreaking, their findings provide valuable clarification in the ongoing scientific debate and thus merit publication.

The submitted article follows the same structure as the CMAJ paper, as it is a refined continuation of their previous work. However, one notable omission is the lack of reference to other research on assortative mixing and vaccinations during the COVID-19 pandemic in the Discussion section. Notably, other researchers have also explored this topic, as evidenced by the following studies:

- Hiraoka, Rizi, Kivelä, Saramäki. ‘Herd Immunity and Epidemic Size in Networks with Vaccination Homophily’. Physical Review E 105, no. 5 (12 May 2022): L052301. https://doi.org/10.1103/PhysRevE.105.L052301.

- Burgio, Steinegger, Arenas. ‘Homophily Impacts the Success of Vaccine Roll-Outs’. Communications Physics 5, no. 1 (28 March 2022): 1–7. https://doi.org/10.1038/s42005-022-00849-8.

- Watanabe & Hasegawa. ‘Impact of Assortative Mixing by Mask-Wearing on the Propagation of Epidemics in Networks’. Physica A: Statistical Mechanics and Its Applications 603 (1 October 2022): 127760. https://doi.org/10.1016/j.physa.2022.127760.

It is worth noting that these papers were published in traditionally non-medical journals.

6. PLOS authors have the option to publish the peer review history of their article (what does this mean?). If published, this will include your full peer review and any attached files.

Reviewer #1: No

Reviewer #2: No

---

## [Editor Report · Decision Letter 1]

27 Dec 2023

Impact of Immune Evasion, Waning and Boosting on Dynamics of Population Mixing Between a Vaccinated Majority and Unvaccinated Minority

PONE-D-23-24559R1

Dear Dr. Fisman,

We’re pleased to inform you that your manuscript has been judged scientifically suitable for publication and will be formally accepted for publication once it meets all outstanding technical requirements.

Kind regards,

Shinya Tsuzuki, MD, PhD

Academic Editor

PLOS ONE

Additional Editor Comments (optional):

I feel the authors responded appropriately to the concerns raised by the reviewers.
---

## [Editor Report · Acceptance letter]

21 Feb 2024

PONE-D-23-24559R1 

PLOS ONE

Dear Dr. Fisman, 

I'm pleased to inform you that your manuscript has been deemed suitable for publication in PLOS ONE. Congratulations! Your manuscript is now being handed over to our production team.

Kind regards, 

on behalf of

Dr. Shinya Tsuzuki 

Academic Editor

PLOS ONE